# Generating Adversarial Examples for Robust Deception against Image Transfer and Reloading

Category: Research

### ABSTRACT

Adversarial examples play an irreplaceable role in evaluating deep learning models' security and robustness. It is necessary and important to understand the effectiveness of adversarial examples to utilize them for model improvement. In this paper, we explore the impact of input transformation on adversarial examples. First, we discover a new phenomenon. Reloading an adversarial example from the disk or transferring it to another platform can deactivate its malicious functionality. The reason is that reloading or transferring images can reduce the pixel precision, which will counter the perturbation added by the adversary. We validate this finding on different mainstream adversarial attacks. Second, we propose a novel Confidence Iteration method, which can generate more robust adversarial examples. The key idea is to set the confidence threshold and add the pixel loss caused by image reloading or transferring into the calculation. We integrate our solution with different existing adversarial approaches. Experiments indicate that such integration can significantly increase the success rate of adversarial attacks.

**Keywords:** Adversarial Examples ,Robustness, Reloading

**Index Terms:** Computing methdologies—Computer vision problems; Neural networks—Security and privacy—Software and application security;

## 1 INTRODUCTION

DNNs are well known to be vulnerable to adversarial attacks [1]. The adversarial algorithm can add small but carefully crafted perturbations to an input, which can mislead the DNN to give incorrect output with high confidence. Extensive work has been done towards attacking supervised DNN applications across various domains such as image [2–5],audio [6–8], and natural language processing [9, 10]. Since DNNs are widely adopted in different AI tasks, the adversarial attacks can bring significant security threats and damage our everyday lives. Moreover, researchers have demonstrated the possibility of adversarial attacks in the physical world [11, 12], proving that the attacks are realistic and severe.

In addition to attacking DNN models, generating powerful and robust adversarial examples also has very positive meanings. First, adversarial examples can be used to test and evaluate the robustness and security of DNN models. The more sophisticated and stealthy the adversarial examples are, the more convincing their evaluation results will be. Second, generating adversarial examples can also help defeat such adversarial attacks. One promising defense is adversarial training [2], where adversarial examples will be included in the training dataset to train a model that is resistant to those adversarial examples. Obviously, if we inject more powerful adversarial examples into the training set, the model will be more robust.

In this paper, we explore and evaluate the effectiveness of adversarial examples with transformation. Guo et al. [13] studied the image transformation (cropping-scaling, bit-depth reduction, compression) as a defense against adversarial attacks; Dziugaite et al. [14] conducted comprehensive evaluations on the effectiveness of adversarial examples with JPG compression. Unlike the above work that *actively* transforms the images, we consider cases where images are *passively* transformed due to reloading or transferring. We discover that an image will lose certain precision when it is reloaded from the disk, or transferred to a different platform. Such precision

reduction in an adversarial example can counter the adversarial perturbation, making the attack ineffective. We evaluate adversarial examples' effectiveness with different mainstream methods and find that most of the methods will fail after the image is reloaded or transferred.

To generate robust adversarial examples against image reloading or transferring, we propose a novel approach, Confidence Iteration (CI). Generally, our CI approach dynamically checks the generated examples' confidence score to evaluate its effectiveness after being reloaded or transferred. By doing so it can filter out the less qualified adversarial examples.

Our approach has several advantages. First, it is generic and can be integrated with existing adversarial attacks for enhancement because it can be called outside of the adversarial algorithm. Second, the adversarial examples generated by our approach have higher success rates before and after they are reloaded or transferred. Third, the adversarial examples generated by our approach have a lower detection rate by state-of-the-art defense solutions. We expect that our solution can help researchers better understand, evaluate and improve DNN models' resistance against various adversarial examples.

In summary, we make the following contributions:

- we are the first to find that the adversarial examples can be ineffective after being reloaded or transferred. We confirm our findings through comprehensive evaluations;

- We propose an effective method, Confidence Iteration, to generate more robust adversarial examples, which can maintain high attack performance under image transformation.

The rest of the paper is organized as follows: Section 2 gives the background and related work about adversarial attacks and defenses. Section 3 describes and evaluates the adversarial examples' effectiveness after image reloading and transferring. We introduce our approach in Section 4 and evaluate it in Section 5. Section 6 concludes the paper.

## 2 RELATED WORKS

In this section, we give a brief background about attack and defense techniques of adversarial examples. We also introduct the resistance of adversarial examples against input transformation.

### 2.1 Adversarial Attack Techniques

An adversary carefully crafts adversarial examples by adding imperceptible and human unnoticeable modifications to the original clean input. The target model will then predict this adversarial example as one attacker-specific label (targeted attack), or arbitrary incorrect labels (untargeted attack). Most adversarial attacks require that the $L_p$ norm of the added modifications cannot exceed a threshold parameter $\varepsilon$. Different adversarial attack techniques have been proposed. We will describe six common attack methods below.

**Fast Gradient Sign Method (FGSM) [2].** The intuition of FGSM is that the adversary can modify the input such that the change direction is completely consistent with the change direction of the gradient, making the loss function increase at the fastest speed. Such changes can cause the greatest impact on the classification results, making the neural network misclassify the modified input.

**Basic Iterative Method (BIM) [15].** This is a simple extension of FGSM. The basic idea of BIM is to apply FGSM for several iterations, with a small step size for each iteration. The number of iterations is determined by $min(\varepsilon+4, 1.25\varepsilon)$.

**DeepFool [16].** Deepfool is based on the assumption that models are fully linear. There is a polyhedron that can separate individual classes. The DeepFool attack searches for adversarial examples with minimal perturbations within a specific region using the L2 distance. Therefore, one big advantage of DeepFool is that it can automatically determine the optimal perturbation threshold $\varepsilon$.

**Decision-Based Attack [17].** The decision-based attack starts from a large adversarial perturbation and then seeks to reduce the perturbation while staying adversarial. It is a method that only relies on the model's final decision. A perturbation is sampled from a proposal distribution at each step, which reduces the distance of the perturbed image towards the original input. They find progressively smaller adversarial perturbations according to a given adversarial criterion. The decision-based attack finally generates an adversarial example with little disturbance near the classification boundary.

**HopSkipJump Attack [18].** HopSkipJump Attack is an algorithm based on a novel estimate of the gradient direction using binary information at the decision boundary. Different from Decision-Based Attacks, which need a large number of model queries, Hop-SkipJump Attack requires significantly fewer model queries and generation time. What is more, in HopSkipJump Attack, the perturbations are used to estimate a gradient direction to handle the inefficiency in Boundary Attack.

**Projected Gradient Descent(PGD) [19].** Their PGD attack consists of initializing the search for an adversarial example at a random point within the allowed norm ball, then running several iterations of the basic iterative method [15] to find an adversarial example.

### 2.2 Adversarial Example Defense Techniques.

Existing approaches for defeating adversarial examples mainly fall into two categories, as described below.

**Adversarial Training.** Szegedy et al. [2] proposed that by training the neural network with the mixed dataset of adversarial examples and original clean samples, the new model will be resistant to adversarial examples. However, Moosavi-Dezfooli et al. [20] showed that an adversary can still generate new examples to fool the defense model.

**Adversarial Example Detection.** Instead of enhancing the models, these approaches aim to detect adversarial examples. One typical solution is de-noising. Mustafa A et al. [21] proposed the wavelet reconstruction algorithm to map the adversarial examples outside of the manifold region to the natural images' manifold region through a deep image reconstruction network. It can restore the normal discriminability of the classifier effectively. Hinton et al. [22] adopted this reconstruction process of capsule network to detect adversarial examples automatically.

### 2.3 Transformation and Distortion of Adversarial Examples.

Most neural networks trained for image classification are trained on images that have undergone JPG compression, containing the original data subspace.

Dziugaite et al. [14] find that perturbations of natural images (by adding scaled white noise or randomly corrupting a small number of pixels) are almost certain to move an image out of the JPG subspace and, therefore, out of the data subspace. Adversarial examples can, therefore, induce the classification network to give wrong classification results. However, when the degree of disturbance is small, the pixel disturbance value superimposed on the original image by the adversarial example is also small, which means that these disturbance values are not robust to image compression, storage, and transmission. The pixel loss is the reason why image transformation

or distortion can defeat adversarial examples.Obviously, how to keep pixel perturbation is the solution to this problem.

## 3 Transferring and Reloading of Adversarial Examples

We study different popular image formats and approaches of adversarial attacks and conclude that image transferring and reloading can significantly reduce adversarial attacks' success rate.

### 3.1 Root Cause

There are two reasons that image transferring and reloading can deactivate adversarial examples. First, in an adversarial image generated using existing approaches, each pixel is usually represented as a `float` value. When we store the image into the disk, the pixels will be converted into `int` type to save space. Such accuracy loss can make the adversarial example ineffective when we reload it from the disk. We find that the mainstream image formats (BMP, JPEG, and PNG) all perform such pixel conversion. Second, when we transfer an image to a different platform via networks, the image is usually compressed to save the network traffic. For instance, we use the WeChat application to send pictures from a smartphone to a laptop and find that the application will compress the pictures with an 80% compression rate by default.

Although such conversion and compression types have a human unnoticeable impact on the images, they can significantly affect adversarial attacks' success rate. The adversary's goal is to find the *smallest* perturbation that causes the model to classify the image into an attack-specific category. Common techniques usually move the original clean samples towards the classification boundary and stop when the samples just cross the boundary to make sure that the added perturbation is small. So the adversarial examples have very high precision requirements for their pixel values. The small changes caused by image reloading or transferring can move the adversarial images to classes different from the one the adversary desires, making the adversarial examples ineffective. Here, we use Figure 1 to directly illustrate the adverse effects of image reloading and image format transformation on the adversarial effect of the adversarial example. Below we conduct a set of experiments to validate those effects empirically.

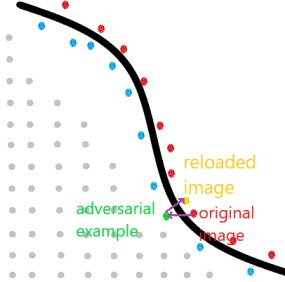

Figure 1: Red dots represent data, and the gray line represents the hyperplane that can separate individual classes. The gray dots represent the inner boundary of the adversarial examples. The green dot represents a specific adversarial example. The yellow dot represents that reloading can project this adversarial example back into the original sample space.

### 3.2 Experiments

#### 3.2.1 Impact of image reloading.

We first empirically check the effectiveness of adversarial examples after being saved and reloaded.

| 31.6 | 70.4 | 48.9 | | 31.0 | 70.0 | 48.0 | | 32.0 | 70.0 | 49.0 | | 32.0 | 70.0 | 49.0 |
|------|------|------|---|------|------|------|---|------|------|------|---|------|------|------|
| 28.1 | 70.6 | 47.1 | | 28.0 | 70.0 | 47.0 | | 28.0 | 71.0 | 47.0 | | 28.0 | 71.0 | 47.0 |
| 23.1 | 67.5 | 44.5 | | 23.0 | 67.0 | 44.0 | | 23.0 | 68.0 | 45.0 | | 23.0 | 68.0 | 45.0 |
| (a) Original | | | | (b) JPG | | | | (c) PNG | | | | (d) BMP | | |

Figure 2: Pixel values before and after saving/reloading

**Precision loss.** We generate a 3×3 image, and add each pixel value with a random perturbation $q$ between 0 and 1. Then we save the image into three different formats (JPG, BMP, PNG) and then reload it into the memory. All the operations are done under windows10.

Figure 2 shows the pixel values of the original image (2a) and reloaded JPG (2b), PNG (2c) and BMP (2d) images, respectively. We observe that each image format has precision loss due to the type conversion from `float` to `int`: JPG format directly discards the decimals. In contrast, PNG and BMP formats round off the decimals. Although such estimation does not cause visual-perceptible effects to the image, it can affect the results of adversarial attacks, as these attacks require precise pixel-level perturbations. We demonstrate the effects below.

**Effectiveness of adversarial examples.** We measure the performance of adversarial examples after being reloaded or transferred. We select six commonly used approaches of adversarial attacks: Decision-Based Attack [17], HopSkipJump Attack [18], Deepfool [16], BIM [15], FGSM [2] ,and PGD [19]. For each approach, we generate some adversarial examples. Decision-Based Attack, HopSkipJump Attack and PGD use ResNet50 classifier. Deepfool uses ResNet34 classifier. BIM and FGSM use the VGG11 classifier. Furthermore, all adversarial examples are tested with the classifier used at the time of generation.

We find that all the six adversarial attack methods measure the classification number and confidence of adversarial examples at the time of generation to judge whether the adversarial attack is successful. In fact, the classification number and confidence at this time are not true, because the model does not classify the real image at this time. They all use models(for example, ResNet50) to classify the generated Numpy array instead of the real picture itself. It means, so far, they have not generated the image form of the adversarial examples. To test the effectiveness of the adversarial examples, we use `cv2.imwrite` and `plt.savefig` to download the adversarial examples locally. Next, we use the same model(for example, ResNet50) to load the adversarial examples saved locally. In this paper, we refer to the above behavior as "Reloading."

We also find that when images are transmitted through instant messaging software, companies compress them to save bandwidth, which results in a loss of pixels in the image, which is detrimental to the adversarial examples generated by subtle perturbations. For example, when we use WeChat to send an image to a friend, our friend can see the compressed image with only a small amount of traffic. Instead of clicking the "download the original" button, we save the compressed image locally and use the above model to categorize it. The above process is referred to as "Transferring" in this paper.

We use Figure 3 and Figure 4 to illustrate adversarial examples' confidence values after being reloaded and transferred. Different colors represent different classification Numbers, the height of the column represents confidence, and each block represents six algorithms from left to the right: Decision-Based Attack [17],HopSkipJump Attack [18],DeepFool [16],BIM [15],FGSM [2], and PGD [19]. We can see that the initial image can be correctly classified with high confidence in all six algorithms. Besides, all the adversarial examples generated by the algorithms can be classified into other categories, which means that the six algorithms' adversarial examples have the

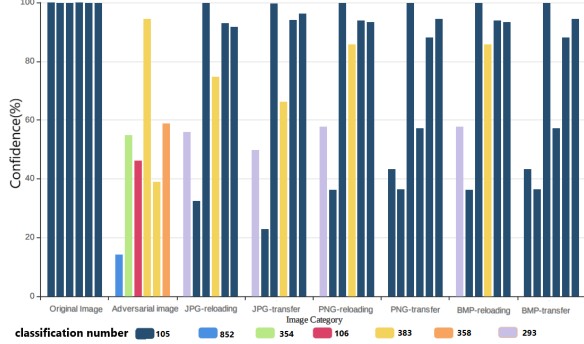

Figure 3: Classification number and confidence of adversarial examples after being reloaded and transferred.

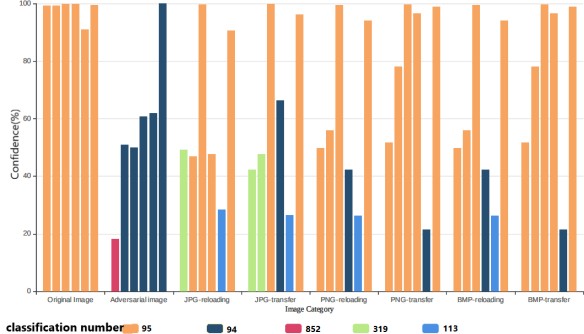

Figure 4: Classification number and confidence of adversarial examples after being reloaded and transferred for another picture.

adversarial ability to deceive classification models into giving false results.

Surprisingly enough, we find that regardless of adversarial examples are saved in JPG, PNG, or BMP, most of them could be classified as the original clean image when they are reloaded or transferred. Some even had high confidence. As reflected in the image, the image after Reloading or Transferring is classified as the original clean image with the same color.

We hope to use more straightforward data to show you this phenomenon. As a result, Table 1 and Table 2 are two experimental results of another two groups of Reloading and Transferring. The data in the table represents the classification number and confidence (data in brackets). We can find that many of the adversarial examples generated by the six kinds of adversarial attacks cannot maintain their attack ability after being reloaded or transferred. After being reloaded or transferred, the adversarial examples will be classified as original clean samples' labels (such as 90 and 129) by the classifier. In order to verify that the adversarial examples with high confidence also have Reloading and Transferring problems, we conduct the following experiments with results in Table 3:

We can find that the adversarial examples of Picture1~Picture4 with high confidence as shown in Figure 5, after being reloaded or transferred, a large part of them are classified as original clean samples in Table 3, proving that the adversarial examples with high confidence also have Reloading and Transferring problems.

All data in Tables 1 to 3 are all derived from the ResNet50 model.

**Cross Validation.** Instead of using the same model to verify adversarial examples' effectiveness, we conduct two sets of cross-validation experiments. One set uses the Reloaded images, and the other uses the Transferred images. The classification number of the initial clean image is 129. The classification numbers of their adversarial examples generated by the six adversarial algorithms are no

longer 129, which means that the adversarial attack is successful(not shown in Table 4). We feed the two sets of adversarial examples generated by algorithm A into algorithm B after they are Reloaded or Transferred, to cross-verify the adversarial effectiveness of adversarial examples after being Reloaded or Transferred. Table 4 shows their classification Numbers and Confidence in other algorithms.

Obviously, no matter after Reloaded or Transferred, the adversarial examples lose their effectiveness in their own and other adversarial algorithms. After WeChat transmission, due to the existence of image compression during the transmission process, four new items in the table are classified to be recovered as clean samples.

**Multiple attacks** In this section, we use the existing adversarial examples as input and conduct other adversarial attacks. The new adversarial examples after reloaded are tested for effectiveness. The results are shown in Table 5.

As shown in Table 5, even if we send the generated adversarial examples into another generation algorithm again, the problem that reloading and transferring results in the decrease of the adversarial effectiveness also exists and is very serious. In Table 5, we see that in addition to an item that failed to generate the adversarial example across models and an item misclassified as classification number 533, other adversarial examples are all classified as the initial clean sample's classification number 129.

The above chart synoptically shows that Reloading and Transferring will significantly reduce the effectiveness of the adversarial attack. This is true for single attacks, cross attacks, and multiple attacks.

### 3.3 Spectrum Analysis.

Next, spectrum analysis is performed on the adversarial examples used in Table 1 and Table 2.

The spectrum analysis results are shown in Figure 6. From left to right are the initial images, adversarial examples generated by BIM,FGSM and Deepfool algorithms. We can find that the Deepfool algorithm can retain the original clean sample's original appearance to the greatest extent. In contrast, FGSM algorithm introduces more noise points, which is reflected in the spectrum map, that is, FGSM algorithm generates a more uniform distribution of the spectrum map with more low-frequency components. This is why the adversarial examples generated by the FGSM algorithm have better resistance to Reloading and Transferring loss in Table 1 and Table 2.

The results of the wavelet transform spectrum diagram of the original picture and adversarial examples of BIM, FGSM, and Deepfool are shown in Figure 7 from left to right. Obviously, in the wavelet domain, the original clean image is closest to the adversarial example generated by Deepfool, both in low and high-frequency components, which means that Deepfool's algorithm can counter the attack with minimal perturbation and is least likely to maintain its antagonism at the same time. FGSM algorithm exerts a large disturbance, so the high and low-frequency components in the wavelet domain are quite different from the original clean image, maintaining the antagonism relatively well.

### 4 A ROBUST APPROACH TO GENERATING ADVERSARIAL EXAMPLES

As discussed in Section 3, adversarial examples generated from existing techniques will become ineffective after being reloaded or transferred. In this section, we propose an efficient and robust approach, Confidence Iteration (CI), to produce adversarial examples that are resistant to the processes of Reloading or Transferring. CI is generic: it can be integrated with all existing adversarial example techniques to improve their robustness while maintaining their advantages.

Our CI approach's intuition is that an adversarial example's confidence score of the attacker-specific classification number reflects this example's resistance against input reloading or transferring. We use

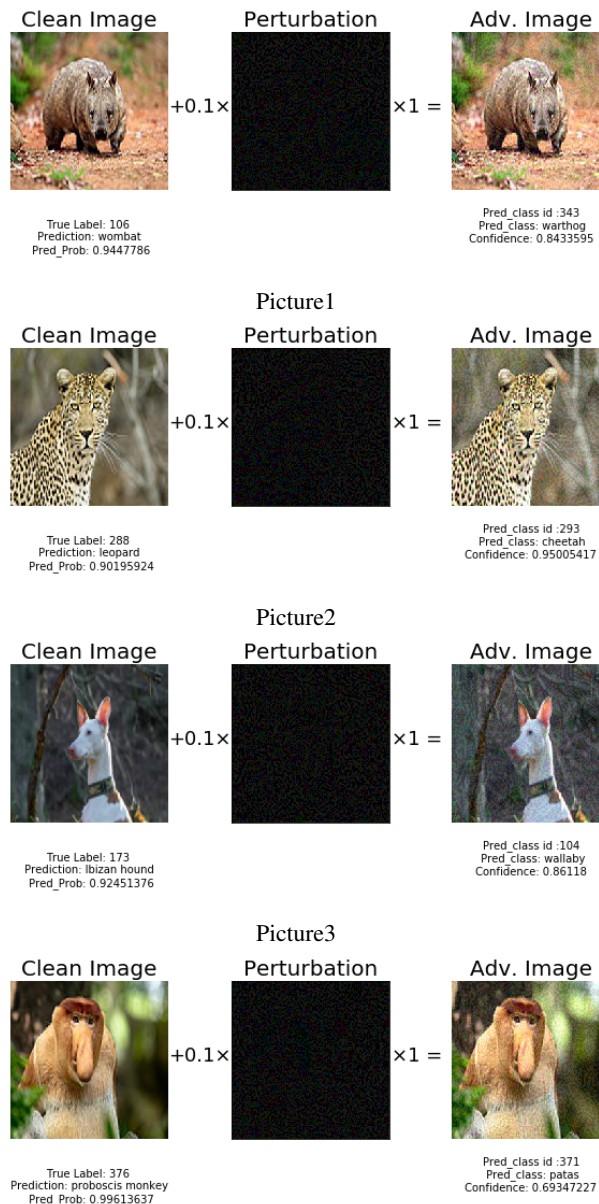

Figure 5: Adversarial Examples generated from Picture1∼Picture4

Table 1: Classification number and confidence of an adversarial example after being reloaded and transferred

| Classification number(confidence) | | Attack Decision | HopSkipJump | Deepfool | BIM | FGSM | PGD |
|---|---|---|---|---|---|---|---|
| Original images | | 90(74.060%) | 90(74.060%) | 90(99.811%) | 90(99.582%) | 90(97.312%) | 90(100.000%) |
| adversarial images | | 852(15.062%) | 84(48.441%) | 95(49.315%) | 95(61.163%) | 735(44.672%) | 318(100.000%) |
| JPG | reloading | 90(72.291%) | 90(69.921%) | 90(52.677%) | 90(46.958%) | 84(99.217%) | 90(99.651%) |
| | transferring | 90(63.671%) | 90(93.686%) | 90(52.985%) | 90(47.276%) | 84(99.402%) | 90(96.650%) |
| PNG | reloading | 84(52.540%) | 84(83.981%) | 90(43.454%) | 90(45.934%) | 84(99.421%) | 90(94.402%) |
| | transferring | 90(82.835%) | 90(50.656%) | 90(80.671%) | 90(36.895%) | 84(89.627%) | 90(99.985%) |
| BMP | reloading | 84(52.540%) | 84(83.981%) | 90(43.454%) | 90(45.934%) | 84(99.421%) | 90(94.402%) |
| | transferring | 90(82.835%) | 90(50.656%) | 90(80.671%) | 90(36.895%) | 84(89.627%) | 90(99.985%) |

Table 2: Classification number and confidence of another adversarial example after being reloaded and transferred

| Classification number(confidence) | | Attack Decision | HopSkipJump | Deepfool | BIM | FGSM | PGD |
|---|---|---|---|---|---|---|---|
| Original images | | 129(89.531%) | 129(89.531%) | 129(86.374%) | 129(71.917%) | 129(91.494%) | 129(98.182%) |
| adversarial images | | 852(12.363%) | 132(36.282%) | 128(48.604%) | 128(98.746%) | 915(5.642%) | 128(97.858%) |
| JPG | reloading | 132(35.742%) | 129(65.183%) | 129(60.726%) | 129(87.825%) | 132(51.324%) | 129(81.000%) |
| | transferring | 132(34.461%) | 129(58.947%) | 129(88.792%) | 129(85.496%) | 132(30.130%) | 129(98.601%) |
| PNG | reloading | 132(53.513%) | 129(64.022%) | 129(53.670%) | 129(85.081%) | 132(53.185%) | 128(38.533%) |
| | transferring | 129(36.472%) | 129(77.169%) | 129(81.671%) | 129(81.244%) | 129(41.192%) | 129(89.833%) |
| BMP | reloading | 132(53.513%) | 129(64.022%) | 129(53.670%) | 129(85.081%) | 132(53.185%) | 128(38.533%) |
| | transferring | 129(36.472%) | 129(77.169%) | 129(81.671%) | 129(81.244%) | 129(41.192%) | 129(89.833%) |

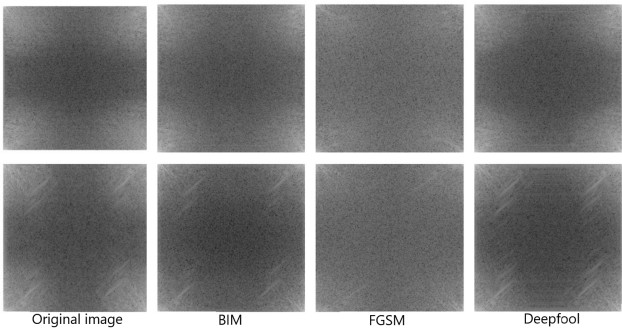

Figure 6: Spectrum analysis of pictures in Table 1 and Table 2

one existing technique (e.g., FGSM, BIM) to generate an adversarial example and save it in the disk locally, and measure its confidence score of the target class. (This actually involves reloading the image.) If the confidence score is higher than a threshold, we will accept this image. Otherwise, we continue to iterate, save it locally (or transform it through WeChat transmission), and measure the target class's reloading confidence score until it meets the confidence requirement or exceeds the iteration number threshold. When the confidence value $c$ meets the expected requirement $p$, the adversarial example image saved to the hard disk at this time has some resistance to the pixel value's variant. Besides, multiple gradient rise caused by multiple iterations will keep the pixel values change with consistent direction. That is to say, after many iterations, the fractional parts of some pixel values will be promoted to the integer part, can no longer be discarded. To measure if an adversarial example is effective after image distortion, we adopt the wavelet reconstruction algorithm [21]. As the name implies, we first process adversarial examples through the wavelet denoising algorithm. Then, we send the denoised image into ESRGAN, A super-resolution reconstructed network. Some adversarial examples with weak attack ability will be classified as initial clean samples after being processed by this algorithm, which means that their attack ability has been lost. By detecting the adversarial examples processed by the wavelet reconstruction algorithm, we could measure the generated adversarial examples' robustness

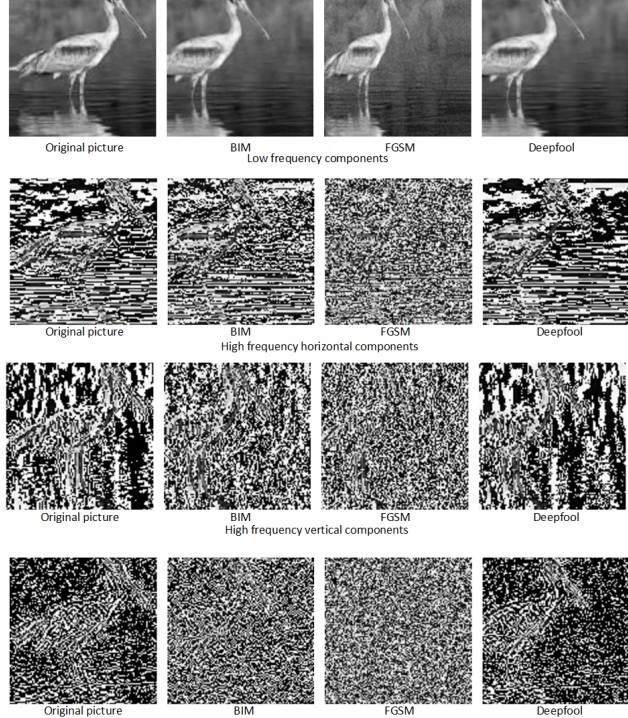

Figure 7: Wavelet transform spectrum diagram of original picture and adversaral examples of BIM, FGSM and Deepfool from left to right

Table 3: Classification number and confidence of adversarial examples generated from Picture1~Picture4 after being reloaded and transferred

| FGSM | | Picture1 | Picture2 | Picture3 | Picture4 |
|------|------|----------|----------|----------|----------|
| Original images | | 106(94.478%) | 288(90.196%) | 173(92.451%) | 376(99.613%) |
| adversarial images | | 343(84.336%) | 293(95.005%) | 104(86.118%) | 371(69.347%) |
| JPG | reloading | 106(99.904%) | 288(49.574%) | 104(28.730%) | 371(34.062%) |
| | transferring | 106(99.953%) | 288(54.895%) | 104(31.623%) | 371(33.070%) |
| PNG | reloading | 106(99.685%) | 608(26.309%) | 173(49.878%) | 376(36.097%) |
| | transferring | 106(99.807%) | 390(47.548%) | 173(47.880%) | 371(66.135%) |
| BMP | reloading | 106(99.685%) | 608(26.309%) | 173(49.878%) | 376(36.097%) |
| | transferring | 106(99.807%) | 390(47.548%) | 173(47.880%) | 371(66.135%) |

Table 4: Classification number and confidence of adversarial examples after being reloaded and transferred using Cross-Validation

| Classification number(confidence) | | Original clean image | Deepfool | BIM | FGSM |
|-----------------------------------|------|----------------------|----------|-----|------|
| Deepfool | reloading | 129(89.16%) | 129(72.14%) | 129(86.31%) | 128(57.74%) |
| | transferring | 129(91.25%) | 128(77.49%) | 129(89.12%) | 129(67.91%) |
| BIM | reloading | 128(72.14%) | 128(77.30%) | 129(60.48%) | 129(65.53%) |
| | transferring | 129(91.81%) | 128(78.98%) | 141(47.95%) | 129(85.49%) |
| FGSM | reloading | 132(82.26%) | 129(40.96%) | 129(14.19%) | 129(58.89%) |
| | transferring | 129(65.64%) | 129(42.91%) | 129(15.98%) | 129(88.35%) |
| PGD | reloading | 129(60.72%) | 129(87.82%) | 129(89.18%) | 129(81.00%) |
| | transferring | 129(66.12%) | 129(68.06%) | 129(89.11%) | 129(98.60%) |

and effectiveness.

---

**Algorithm 1** Confidence Iteration

**Input:** A classifier $f$ with loss function $J$;a real example $\mathbf{x}$ and ground-truth label $y$;

**Input:** The size of perturbation $\varepsilon$;iterations limit number $T_{max}$ and confidence limit value $p$;

**Output:** iterations number $T$;An adversarial example $\mathbf{x}^*$ with $\|\mathbf{x}^* - \mathbf{x}\|_\infty \leq T\varepsilon$

1: $T = 0$;
2: $\mathbf{x}_T^* = \mathbf{x}$;
3: Save $\mathbf{x}_T^*$ as a picture $\mathbf{x}_T^{real}$ on your local hard drive (or transform it through WeChat transmission)
4: Input $\mathbf{x}_T^{real}$ to $f$ and obtain the confidence $c$ and the gradient $\nabla_{\mathbf{x}} J\left(\mathbf{x}_T^{real}, y_{true}\right)$ ;
5: **while** $(T \leq T_{max})$ and $(c \leq p)$ **do**
6:      $\mathbf{x}^* = \mathbf{x}_T^{real} + \varepsilon \cdot \nabla_{\mathbf{x}} J\left(\mathbf{x}_T^{real}, y_{true}\right)$;
7:      $T = T + 1$;
8:      $\mathbf{x}_T^* = \mathbf{x}^*$;
9:      Save $\mathbf{x}_T^*$ as a picture $\mathbf{x}_T^{real}$ on your local hard drive (or transform it through WeChat transmission)
10:      Reload $\mathbf{x}_T^{real}$ to $f$ and obtain the confidence $c$ and the gradient $\nabla_{\mathbf{x}} J\left(\mathbf{x}_T^{real}, y_{true}\right)$ ;
11: **end while**

---

Algorithm 1 summarizes our CI approach. We first input the clean image, generate its adversarial example, and then save the adversarial example locally. The local adversarial example is then reloaded into the classification model and judges whether the adversarial attack can succeed. On the premise of the adversarial attack's success, we give the confidence value $c$ of the adversarial attack, which is obtained by reloading the adversarial example in the hard disk into the classification model. Then we compare the expected confidence threshold $p$ with the current confidence $c$. If the current confidence $c$ is less than the expected confidence threshold $p$ and the current iteration number $T$ is less than the iteration number threshold $T_{max}$. We will run the Confidence Iteration algorithm, save the generated adversarial example locally, and compare c and p. The whole process will not stop until $c$ is greater than $p$ or $T$ equals $T_{max}$.

It is precisely because the CI algorithm has a download, reload, and confidence judgment process. We can apply it to the backend of any adversarial example generation algorithm to enhance the adversarial example's robustness against reloading and transferring.

## 5 EVALUATION

In this section, we conduct experiments to validate the effectiveness of our proposed approach.

### 5.1 Configurations

**Dataset.** To better reflect the real-world setting, we implement a crawler to crawl some images from websites instead of using existing image datasets. We consider the Inception v3 model [23] and restrict the scope of crawled images to the categories recognized by this model. We filter out the crawled images that the Inception v3 model cannot correctly recognize and finally establish a dataset consisting of around 1300 clean images with the correct labels.

**Implementations.** We consider two adversarial example techniques: FGSM and BIM. Our CI approach is generic and can be applied to other techniques as well. We select VGG11 [24] as the target model. We implement these techniques with the CI algorithm using the PyTorch library. We set the iteration upper limit $T_{max}$ as 6, and the confidence threshold $p$ as 70%.

**Metrics.** We adopt two metrics to evaluate the effectiveness of adversarial examples: (1) success rate is defined in Equation 1a. $N_s$ is the number of adversarial examples which can be misclassified by the target model $f$ while its clean images can be classified truly, and

Table 5: Classification number and confidence of adversarial examples after multiple attacks

| A image with classification number 129 | +0 | +Deepfool | +BIM | +FGSM | +PGD |
|---|---|---|---|---|---|
| Deepfool | 129(60.72%) | 129(71.88%) | 129(95.91%) | Unsuccessful generation | 129(92.27%) |
| BIM | 129(89.82%) | 129(92.32%) | 129(99.37%) | 129(98.65%) | 129(90.22%) |
| FGSM | 129(89.18%) | 129(72.24%) | 533(31.53%) | 129(71.53%) | 129(55.72%) |
| PGD | 129(81.00%) | 129(82.52%) | 129(88.24%) | 129(99.71%) | 129(55.72%) |

$N_f$ is the number of adversarial examples that can be predicted as corresponding clean image's label; (2) average confidence score is defined in Equation 1b. $p_i$ is the confidence score from the target model with the highest false classification confidence. (Here we do not consider the adversarial example that can be classified as its clean sample's label by the model.)

$$P_{adv} = \frac{N_s}{N_s + N_f} \tag{1a}$$

$$C_{ave} = \frac{1}{N_s} \sum_{i=1}^{N_s} p_i \tag{1b}$$

## 5.2 Results and Analysis

**Adversarial example generation.** We first show the generated adversarial examples using FGSM, CI-FGSM, BIM, and CI-BIM, as shown in Figure 8. We can see that similar to FGSM and BIM. Our CI-FGSM and CI-BIM can also produce adversarial examples with imperceptible perturbations that can fool the target deep learning model.

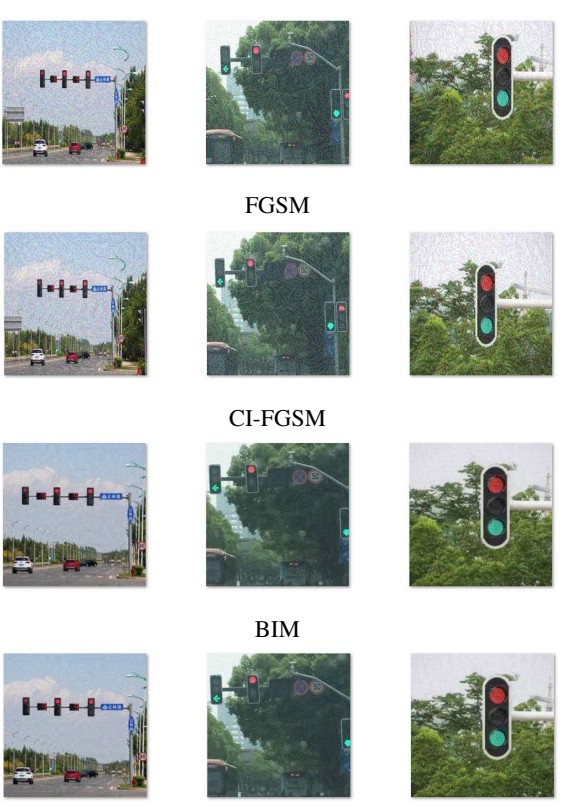

FGSM

CI-FGSM

BIM

CI-BIM

Figure 8: Adversarial examples generated by FGSM, CI-FGSM, BIM, and CI-BIM

**Attack effects after image reloading.** Using different approaches, we generate a large amount of adversarial images, save them to the local disk. Then we reload them and feed them into the target model for prediction. We measure the success rates and average confidence scores of the four algorithms in Table 6. FGSM has a lower success rate and confidence score as it adopts striding perturbation. In contrast, BIM has higher attack performance. For our CI-BIM, although the confidence score is slightly lower than BIM. But the success rate is much higher than that of BIM. Our CI approach is more efficient when $\varepsilon$ is smaller.

Different parameters can lead to different effects of the CI approach. Figure 9 demonstrates the adversarial success rate of adversarial examples from CI algorithm with different threshold $p$. We can see that by increasing $p$, the attack performance can be significantly improved. To boost the adversarial attacks, conventional approaches require setting a large disturbance hyper-parameter $\varepsilon$ (e.g., 16), and large number of iterations $T$ (e.g., 10). To achieve the same effects, our CI approach only needs to increase the threshold while keeping smaller values of $\varepsilon$ ($0.05 \sim 0.2$) and $T$ (e.g., 6) to achieve similar attack effects.

**Resistance against Detection of Adversarial Examples.** In addition to defeating input transformation, our CI-approach is better at evading adversarial example detection. We use the wavelet reconstruction algorithm [21] as the defense method to measure the performance of different adversarial algorithms. After being processed by the wavelet reconstruction algorithm, the adversarial examples with weak attack capabilities will be identified as the initial clean image's label by the classification model. As the name implies, we first process adversarial examples through a wavelet denoising algorithm. Then, we send the denoised image into ESRGAN, A super-resolution reconstructed network. By detecting the adversarial examples processed by the wavelet denoising algorithm, we could measure the generated adversarial examples' robustness. We set the parameter $\varepsilon$ as 0.1 and $\delta$ of the wavelet denoising algorithm from 0.01 to 0.1. Figure 10 shows the comparison results. We can clearly see that although the attack performance of BIM is better than FGSM, the adversarial examples generated by the BIM algorithm are easier to be detected as adversarial examples under the same parameters. On the contrary, our CI method has high attack performance and is not easy to be detected as adversarial examples, especially when the detection parameter $\sigma$ is small.

**Application to other adversarial example techniques.** In addition to BIM, our CI approach can be also applied to other adversarial attack algorithms to boost adversarial examples. Figure 11 shows the attack performance of FGSM with CI and its comparisons with simple FGSM. We can see that the CI approach can improve the attack performance of FGSM, which is more obvious when the parameter $\varepsilon$ is smaller. Simultaneously, the effect of CI-FGSM is much better than that of an ordinary BIM algorithm. CI-BIM algorithm has the best adversarial success rate among the four algorithms, which is also easy to understand. When the parameter $\varepsilon$ is small, the FGSM algorithm uses the small step length for perturbation superposition, BIM algorithm iterates these small step length perturbations, and CI-BIM algorithm iterates again for the iteration of these small step length perturbations on the premise of confidence satisfying the requirements $p$. This is an iteration at different scales. In a sense, our method implements an adaptive step size attack. When the pa-

Table 6: the success rates and average confidence scores of adversarial examples

|  | success rate | | | confidence score | | |
|---|---|---|---|---|---|---|
|  | $\varepsilon$=0.1 | $\varepsilon$=0.2 | $\varepsilon$=0.3 | $\varepsilon$=0.1 | $\varepsilon$=0.2 | $\varepsilon$=0.3 |
| FGSM | 81.4% | 95.8% | 99.2% | 23.3% | 20.3% | 27.0% |
| CI-FGSM | 87.0% | 96.5% | 98.8% | 22.9% | 21.5% | 27.7% |
| BIM | 87.5% | 94.4% | 94.0% | 74.7% | 73.2% | 68.3% |
| CI-BIM | 95.5% | 98.9% | 99.3% | 57.8% | 62.9% | 63.7% |

rameter $\varepsilon$ is relatively small, the adjustment range of the dynamic step length is larger, which means that our CI-BIM algorithm can find adversarial examples with high attack capabilities in a larger range. Essentially, the CI-BIM algorithm has higher adversarial attack performance because of its wider search domain for generating more robust adversarial examples.

## 6 CONCLUSION

In this paper, we evaluate the effectiveness of adversarial examples after being reloaded or transferred. We discover that most mainstream adversarial attacks will fail with such input transformation. Then we propose a new solution, Confidence Iteration, to generate high-quality and robust adversarial examples. This solution can significantly facilitate other existing attacks, increasing the attack success rate and reducing the detection rate. Future work includes more evaluations on the integration with other attack techniques and leveraging Confidence Iteration to enhance DNN models via testing and adversarial training.

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

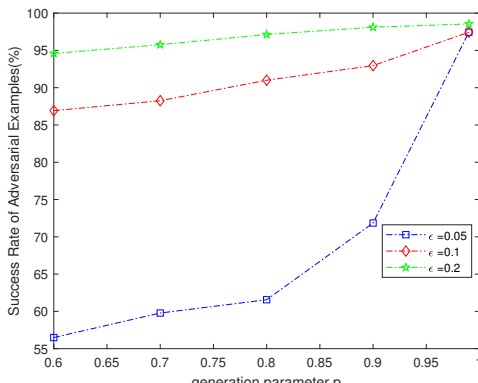

Figure 9: Success rate of the adversarial examples generated by CI approach with different generation parameter *p*

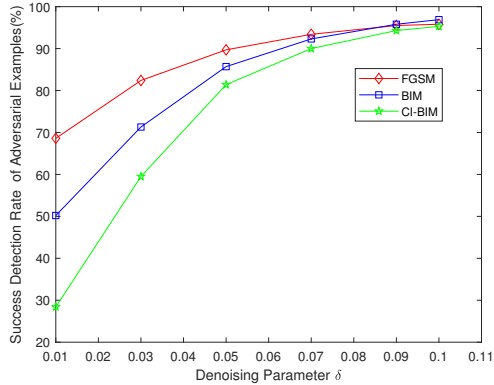

Figure 10: Detection rate of the adversarial examples with wavelet reconstruction algorithm

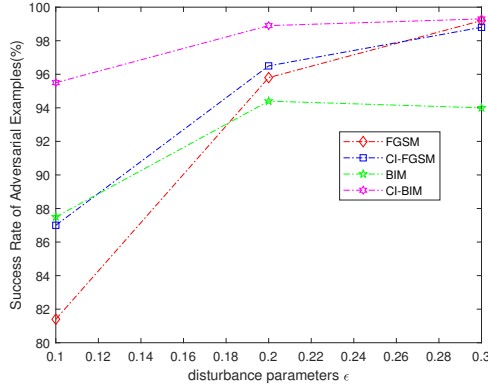

Figure 11: Comparison of the adversarial success rate of FGSM, CI-FGSM, BIM and CI-BIM

