# OpenReview forum: "Generating Adversarial Examples for Robust Deception against Image Transfer and Reloading"
_graphicsinterface.org/Graphics_Interface/2021/Conference — Submitted to GI 2021_

### Official Review · AnonReviewer1 · 2021-01-04
**Review for GI 2021 submission ID: 20.**

**Rating:** 1
**Confidence:** 3

**Review:**

I do not think this work meets the bar for GI 2021.

The main issue is that they way the problem is posed is misleading.  This work over-emphasizes the process of "saving" and "transmitting" while both steps could be invertible. While this might seem like a small detail, it makes the introduction completely incomprehensible (I had no idea what authors mean when they say that saving a piece of data and the re-loading it somehow "degrades" it).

I think it is interesting to study how compression and quantization artifacts affect the quality of adversarial examples. But classifying these artifacts based on the specific formats (JPG, BMP) or social media apps (WeChat) provides little insights and has little scientific value. In the former case, new formats might emerge (or old ones become obsolete) and it would be unclear how findings in this work apply to them. In the latter case, a small change in backend implementation might completely invalidate all the findings at any time.

If authors want to further investigate this with rigor, they could specifically focus on different kinds of compression algorithms and quantization steps that are used in existing formats of data transmission apps. This could be complemented with the study proposed in this paper, to demonstrate how this could translate to practical use case.

As a small note the proposed algorithm: it seems to suggest that physically saving data to the hard drive or transmitting it over the network is an important step. Obviously, it is slow and unnecessary. One can simply use any kind of data format conversion locally to see how this conversion affects the adversity of the proposed example.

It also would be interesting to see if some of these data conversion artifacts could be implemented as differentiable layers in a network. So one could directly optimize for adversarial examples back-propagating over particular compression algorithm.

---

### Official Review · AnonReviewer2 · 2021-01-08
**Generating Adversarial Examples for Robust Deception against Image Transfer and Reloading**

**Rating:** 3
**Confidence:** 3

**Review:**

The authors explore how storing and transmitting images affects the attack performance of adversarial examples. They propose a novel metric called "Confidence Iteration" to generate adversarial examples that are robust to said effects of storage and transmission.

Comments:
- Although the objective is clear, there is no discussion on why more basic techniques cannot be used or fail to address this problem. For example, one would expect that using data augmentation during the training will increase the network's robustness to arbitrary transformations, including photometric, geometric, noise, etc.
- In contrast to the authors' statement, the adversarial examples shown are noticeably noisy. It seems that applying a simple median filter before passing the image to the network would get rid of most of the noise.
- The idea of studying the effects of compression and quantization on adversarial examples' effectiveness is useful. However, I find it trivial and would not call this a discovery of a "new phenomenon". Any information loss (due to any factor, not just these two), which leads to perturbations, can affect the network's performance.
- It is unclear whether the Inceptionv3 model was trained on the created dataset or any other dataset.
- Why was VGG11 used instead of Inceptionv3?
- I find the "Confidence Iteration" trivial. The range of the colour values is not specified. Are the RGB images in the range of [0,1]? If so, the perturbations are up to 30%, which seems quite extreme considering the claim that the adversarial examples are "stealthy".
- The paper requires proofreading as there are a few sentences that do not parse.

---

### Official Review · AnonReviewer3 · 2021-01-13
**Simple and effective method with odd framing**

**Rating:** 6
**Confidence:** 4

**Review:**

The submission presents an algorithm that renders adversarial image examples for neural networks robust against discretisation and compression artefacts.

My main issue with this paper is the framing. The authors claim the observation that adversarial images are generally not robust with respect to image transferring or reloading. It is not clear to me why these specific applications have been picked. I would argue that image compression and quantisation artefacts are obviously a problem for attacks that rely on minimal perturbations of images. It is very clear what it happening here and, contrary to the authors statements, not surprising.

It also seems quite odd that the authors apparently use transfer through WeChat to assess these artefacts instead of just computing compressed images of several levels using any image library. This would be a much cleaner and transparent angle for this paper.

Despite these very general remarks I think the method is valuable for two reasons: The method is simple, yet effective and is easily applicable as a post process to any method generating adversial images.

I would have liked to see a better evaluation to assess the distribution of success rates and confidence values across images. It would also be helpful to focus on specific compression techniques with different parameters.

Since I am not an expert on this topic I can not assess the novelty of the approach or if the issue of discretisation artefacts has been tackled in the context of adversarial image generation before.

---

### Meta-Review · Area_Chair1 · 2021-01-14

**Recommendation:** Reject
**Confidence:** 5

**Metareview:**

The authors present an algorithm that addresses the effects of storing and transmitting images on the quality of adversarial examples. The reviewers find that although it is an interesting concept, this work has significant issues that limit its scientific value. For example, all reviewers find that the claim that compression and quantization effects on the quality of adversarial examples, is a discovery of a new phenomenon, as overstated. At best, this is a trivial and straightforward observation. Additionally, the experiments are limited to two image formats and the application WeChat for transmitting the images.

---

### Decision · Program_Chairs · 2021-01-16

Reject